# Development of Recombinant Immunotoxins for Hairy Cell Leukemia

**DOI:** 10.3390/biom10081140

**Published:** 2020-08-03

**Authors:** Robert J. Kreitman, Ira Pastan

**Affiliations:** 1Laboratory of Molecular Biology, Clinical Center, National Cancer Institute, National Institutes of Health, Bethesda, MD 20892, USA; pastani@mail.nih.gov; 2National Institutes of Health, Building 37/5124b, 9000 Rockville Pike, Bethesda, MD 20892, USA

**Keywords:** hairy cell leukemia, treatment, rituximab, moxetumomab pasudotox, minimal residual disease

## Abstract

Hairy cell leukemia (HCL) is an indolent B-cell malignancy with excellent initial response to purine analogs pentostatin or cladribine, but patients are rarely, if ever, cured. Younger patients will usually need repeat chemotherapy which has declining benefits and increasing toxicities with each course. Targeted therapies directed to the BRAF V600E mutation and Bruton’s tyrosine kinase may be helpful, but rarely eradicate the minimal residual disease (MRD) which will eventually lead to relapse. Moxetumomab pasudotox (Moxe) is an anti-CD22 recombinant immunotoxin, which binds to CD22 on HCL cells and leads to apoptotic cell death after internalization and trafficking of the toxin to the cytosol. Phase I testing achieved a complete remission (CR) rate of 57% in relapsed/refractory HCL. Most CRs were without MRD and eradication of MRD correlated with prolonged CR duration. Patients were often MRD-free after five years. Important mild-moderate toxicities included capillary leak and hemolytic uremic syndromes which could be prevented and managed conservatively. A phase 3 trial met its endpoint of durable CR with acceptable toxicity, leading to FDA approval of Moxe for relapsed/refractory HCL, under the name Lumoxiti. Moxe combined with rituximab is currently being evaluated in relapsed/refractory HCL to improve the rate of MRD-free CR.

## 1. Introduction to Hairy Cell Leukemia

Hairy cell leukemia (HCL) is a B-cell malignancy comprising 2% of leukemias [1,2], amounting to about 1200 new cases per year in the United States [3]. Over 90% of classic HCL cases have the BRAF V600E mutation, which leads to constitutive phosphorylation of ERK and increased proliferation [4,5,6,7]. Other mutations have been reported in classic HCL, in association with BRAF V600E [8,9,10]. Patients often report a history of occupational or residential exposure to chemicals [11,12,13,14], but there is no association with tobacco smoking [15]. Although indolent, the median survival after diagnosis was only four years in 1978 prior to the advent of effective therapies [16]. First-line treatment of HCL underwent a major advance in the late 1980s with the use of purine analogs pentostatin or cladribine, each capable of achieving complete remission (CR) in 76–91% of patients [17,18,19,20,21,22,23]. Extended follow-up showed high 5- and 10-year disease-free survival rates, but without a plateau on the disease-free survival curves, indicating a lack of evidence of cure [24,25]. Although first-line treatment of HCL remains single-agent purine analog [26,27], repeat use is associated with declining CR rates, shorter disease-free intervals, and increasing risk of toxicity with each course, particularly from chronic T-lymphopenia and neuropathy [28,29,30,31]. Repeated treatments are also associated with an increased risk of secondary malignancies [32]. Outcomes using single-agent purine analogs are much poorer with variants of HCL [33,34,35], highlighting the importance of a correct initial diagnosis. 

## 2. Diagnosis of HCL

HCL patients present most commonly with fatigue (80%), and also with poor appetite, fever, weight loss, infections, night sweats, left upper abdominal pain from splenomegaly, and bruising from thrombocytopenia [36]. Patients classically have monocytopenia although sometimes the HCL cells are mistaken for monocytes based on their size [37]. The diagnosis can most specifically and sensitively be made by blood or bone marrow aspirate flow cytometry, which should show bright positivity for CD22, CD20, and CD11c. CD103 is expressed on normal T-cells but is a specific marker for HCL if found on B-cells [36,38]. In classic HCL, CD25 is positive but may be dim to bright, and CD123 is usually positive as well. By bone marrow biopsy immunohistochemistry (IHC), the most common markers include CD20, CD22, and CD19, which also stain normal B-cells, and the more HCL-specific markers tartrate-resistant acid phosphatase (TRAP), DBA44 (CD72), annexin 1A (Anxa1), and the BRAF V600E mutation using the VE1 antibody [4,36,39]. Double staining for Pax5 and either TRAP or CD103 can be highly specific for HCL [40]. The BRAF V600E mutation can also be detected by several PCR techniques including pyrosequencing [7], or allele-specific quantitative PCR [5]. A popular method is digital droplet PCR [41]. Although cytopenias can be observed for years prior to the diagnosis, patients present at a median age of about 55. Three to four times as many patients are male vs. female and HCL is observed more commonly in Caucasians compared with Asians, Arabs, and Africans [15]. DRB1*11, a class II human leukocyte antigen (HLA), is expressed more commonly in HCL compared with the control Caucasian populations, and DRB1*11 is more frequent in Caucasians than in other populations [42]. 

## 3. Differentiation of HCL from Variants

In 1980, Cawley et al. described a variant of HCL where patients had more frequent leukocytosis with less cytopenias including less monocytopenia [43,44]. Now called HCLv, this variant lacks CD25, TRAP, Anxa1, and BRAF V600E, is more aggressive than HCL, and in 2008 was considered a distinct disorder by the World Health Organization [7,43,44]. HCLv differs from HCL in its intra-sinusoidal morphology in bone marrow [35,39,44]. Response of HCLv to single-agent purine analog is poor and requires combinations of purine analog and rituximab [34,45]. Thus, an accurate diagnosis of HCL vs. HCLv is critical. By flow cytometry, CD11c is as bright in HCLv as in classic HCL, but CD25 and usually CD123 are negative [38,44]. CD103 is positive in both HCL and HCLv and its absence requires consideration of splenic marginal zone lymphoma (SMZL) [38,46]. SMZL can be differentiated from HCL and HCLv since SMZL is mainly a white pulp disease [39,44], while HCL/HCLv involve the red pulp of the spleen. Both splenic diffuse red pulp lymphoma (SDRPL), and HCLv are considered “splenic B-cell lymphoma/leukemia, unclassifiable” [35], although SDRPL has less lymphocytosis, anemia, longer overall survival from diagnosis, brighter CD123, and lower CD11c and CD103 expression [35]. A subset of HCLv expresses unmutated immunoglobulin rearrangement IGHV4-34 [33]. IGHV4-34+ HCL can also have an immunophenotype like classic HCL (TRAP+, bright CD25+) but remains aggressive, lacks BRAF V600E, is associated with nodal disease, and responds poorly to first-line purine analog. Thus, IGHV4-34+ HCL/HCLv may be considered a molecularly defined variant overlapping with HCLv [7,33,47,48]. Aberrant markers in classic HCL like CD38 may be associated with shorter time to salvage therapy [49], but other non-HCL markers like CD5 or CD10 may be expressed without prognostic implications [50,51,52,53]. 

## 4. When Treatment Is Indicated in HCL

Although some patients may not need treatment for years or decades, most do and the most common indications for treatment include neutrophils <1–1.5 × 10^9^ cells/L, hemoglobin <10–12 g/dL, and platelets <100 × 10^9^ cells/L [19,21,27,54]. We and others favor the more stringent <1, <10, and <100 limits of these respective criteria given the toxicity of chemotherapy for HCL and the importance of avoiding excessive retreatment [21,55,56,57,58,59,60]. Additional criteria for treatment include malignant lymphocytosis >5 or >20 × 10^9^/L, symptomatic splenomegaly, enlarging lymph nodes >2 cm, and frequent infections. Additional criteria are particularly important for those without cytopenias, like in HCLv or in HCL after splenectomy, since these patients would otherwise not qualify for clinical trials until much more advanced. After prior therapy, it is important to determine whether cytopenias are treatment- or disease-related. An important pitfall to avoid is to perform a bone marrow too early after treatment, and if residual disease is visible administer more therapy. Such a patient might achieve CR without further treatment. For this reason, post-treatment bone marrow biopsies should be delayed at least four–six months after last treatment [27]. Even so, it may still be difficult to determine whether the presence of residual disease and cytopenias indicate the need for additional treatment. In this situation, it is often helpful to obtain peripheral blood flow cytometry at several points in time: an increasing HCL count indicates the cytopenias are due to the residual disease, while undetectable HCL cells in the blood often indicate a good response to last treatment and suggest cytopenias might resolve with additional time.

## 5. Criteria for Response in HCL

By the 1987 Consensus Resolution [61], and the more recent 2017 guidelines [26,27], CR requires elimination of HCL cells by morphologic (non-immunologic) stains, like Wright stain of blood and bone marrow aspirate, and hematoxylin/eosin (H/E) stain of the bone marrow biopsy. Resolution of cytopenias is required for neutrophils ≥1.5 × 10^9^/L, hemoglobin ≥11–12 g/dL, and platelets ≥100 × 10^9^/L [27,55,56,57,58,59,60,61,62]. We typically drop the hemoglobin requirement when iron deficiency is documented, since patients often have limited iron stores pretreatment and iron deficiency may limit rapid recovery of normal red blood cells. Table 1 lists the consensus response criteria along with variations or additions used. Partial response (PR) requires at least a 50% reduction in palpable spleen, liver, and lymph node size, and resolution of cytopenias to levels required by CR. We and others accept at least 50% improvement in neutrophils, hemoglobin, and platelets as sufficient for PR [21,55,56,57,58,59,60,62,63]: a PR with resolution of cytopenias to CR levels has been considered a “good PR” (GPR) [62] or a hematologic remission (HR) [59,60,64,65]. We consider a patient receiving blood transfusions at baseline to have achieved a PR if hemoglobin improved to ≥9.0 g/dL even if the baseline was >6.0 g/dL, provided the improvement occurred ≥4 weeks after last blood transfusion or growth factor [60]. Similarly, resolution of neutropenia should occur ≥4 weeks after last growth factor. A minimum duration of resolved cytopenias was not specified in the guidelines [27], but four weeks is often used [19,59,60,62,64,65,66]. We require four weeks for multiple relapsed HCL protocols, but a single assessment for first- or second line HCL treatment, since improvements are more durable after therapy of early HCL [66,67]. The older PR criteria requiring ≤5% circulating HCL is no longer required for PR [27], although many protocols including ours required ≥50% reduction in peripheral blood HCL cell count [19,21,68]. The PR requirement for ≥50% reduction in HCL infiltration by bone marrow biopsy [61] is one that we avoid due to the heterogenicity and hence inaccuracy of bone marrow biopsy in quantifying marrow infiltration. We also try to limit posttreatment bone marrow procedures to those patients who might qualify for CR, rather than perform them in patients just to document PR. To avoid investigator bias in clinical trials, imaging with CT or MRI is often used instead of physical exam or ultrasound for documenting resolution in adenopathy and splenomegaly. If so, resolution of lymph nodes to ≤2 cm in short axis, and resolution of spleens to either ≤17 cm in diameter or >25% decreased from baseline can be considered consistent with CR. A ≥50% reduction in products of lymph node perpendicular diameters or reduction to a size consistent with CR by CT or MRI is considered consistent with PR. It is notable that residual splenomegaly and hypersplenism causing thrombocytopenia may remain in the absence of residual disease, and splenectomy may not be indicated in these cases [69]. Blood flow cytometry and serum soluble CD25 or CD22 [70] may be helpful in ruling out residual splenic involvement. Residual adenopathy >2cm in short axis can often indicate residual HCL in patients otherwise appearing disease-free, and such patients may require lymph node biopsy to determine response.

## 6. Minimal Residual Disease (MRD)

Since HCL has been studied for over 60 years, it is important to keep the definition for CR relatively constant, which is why CR criteria are limited to morphologic stains. With the advent of immunologic stains, HCL cells not apparent by H/E can readily be seen by immunohistochemistry. Blood and bone marrow flow cytometry using cocktails of antibodies can detect <0.01% HCL cells. MRD criteria by immunohistochemistry include CD20+ or DBA44+ cells mostly consistent with HCL by morphology, and B-cells at least as numerous as T-cells [72]. We have used this older definition of immunohistochemistry MRD [67] since it is most established, but it lacks sensitivity compared with newer techniques like Pax5/CD103 or Pax5/TRAP double staining [40]. One might consider the presence of rare Pax5/CD103 or Pax5/TRAP double-stained cells (6–10 cells on the slide of a 1–2 cm core) to be positive for immunohistochemistry MRD. Additional correlation between immunohistochemistry and flow cytometry is currently being obtained using these double stains. Bone marrow aspirate flow cytometry is most sensitive by far for MRD [34,64,66,67]. A cutoff of ≥0.01% has been used in some studies [64,65], but in others, even suspicious MRD by flow cytometry has been considered positive [73]. Detecting resolution of MRD is particularly relevant for CR achieved by treatments capable of eradicating MRD, and this is a special feature of anti-CD22 recombinant immunotoxin. 

## 7. Introduction to Recombinant Immunotoxins

Recombinant immunotoxins are engineered molecules containing a protein toxin genetically fused to a fragment of a monoclonal antibody [74]. They are like antibody drug conjugates (ADCs), in that they bind to the cell surface and kill the cell after internalization. However, in ADCs, the toxic moiety is a chemotherapeutic agent, which the cancer may be resistant to. In recombinant immunotoxins, the therapeutic payload is a protein toxin which kills by a different mechanism: catalytic inactivation of protein synthesis [75,76,77,78]. Protein synthesis arrest leads to a fall in MCL-1, and activation of BAK and the apoptotic cascade [79].

## 8. Plant vs. Bacterial Toxins

Plant toxins like ricin have been used to make chemical conjugates, including one targeting CD22 [80,81]. However, bacterial toxins like Pseudomonas exotoxin A (PE) and diphtheria toxin (DT) are more convenient for creating chimeric recombinant toxins, since these bacterial toxins are naturally made as single chains [82,83]. Unlike DT, humans do not get vaccinated against PE, which is derived from *Pseudomonas aeruginosa*, and only a minority of patients will have pre-existing antibodies capable of neutralizing PE. Immunotoxins are thought to work by binding to the host cell, internalizing and unfolding in an acidic endocytic vesicle [84], undergoing furin proteolytic cleavage between the ligand and toxin [85,86,87], reducing the disulfide bond holding the cleavage fragments together [88], trafficking of the released ADP-ribosylating fragment to the endoplasmic reticulum [89,90], translocation to the cytosol [91,92], and ADP-ribosylation of EF2 [93,94] causing protein synthesis inhibition and apoptotic cell death [79,95,96,97,98]. 

## 9. LMB-2 Targeting CD25

Figure 1 shows three recombinant immunotoxins tested in patients with hematologic malignancies. The first recombinant immunotoxin produced contains the variable domains of an anti-CD25 monoclonal antibody (Mab) anti-Tac arranged in single-chain Fv form, connected to a 40 kDa truncated form of PE called PE40 [99,100]. Since PE40 is missing the binding domain 1a of PE, anti-Tac(Fv)-PE40 binds to CD25 rather than to normal cells. 

To improve protein folding during production, the connector between the variable domains and toxin was modified. PE40 was shortened to PE38 by removal of amino acids containing a disulfide bond, resulting in anti-Tac(Fv)-PE38 [104] or LMB-2. The anti-CD25 recombinant immunotoxin was cytotoxic toward primary leukemic cells from patients with CD25+ malignancies, including adult T-cell leukemia [101,105] and HCL [106]. In phase 1 testing in both T- and B-cell malignancies, the best results were observed in patients with HCL, with three PRs and one CR out of four patients [55,56]. Its success was based on the high expression of CD25 in most patients with classic HCL [106]. 

## 10. Development of anti-CD22 Recombinant Immunotoxins for HCL

CD22, also called Siglec 2, is expressed mainly by B-lymphocytes and functions to inhibit B-cell receptor signaling [107,108]. The rationale for the development of a CD22 recombinant immunotoxin was based on characteristics of both HCL and non-HCL hematologic malignancies. CD22 is expressed at a much higher level than CD25 in most classic HCL patients [38,106]. HCLv lacks CD25 expression but has high CD22 expression like classic HCL [38]. Other B-cell malignancies usually express CD22 more than CD25 [109]. Compared with CD19, CD22 was found to internalize more rapidly, making it a better target for a recombinant immunotoxin [84]. Finally, CD22 expressing normal B-cells can be replaced from precursor cells not expressing CD22 [110,111]. CD22 was therefore chosen as a target for recombinant immunotoxin development and BL22 was designed and produced [112,113]. BL22 (Figure 1) contains an Fv that binds to CD22, attached to PE38. [103,109,111,114]. BL22 was tested in phase 1 and 2 clinical trials and achieved CR rates of 47–61% in patients with HCL [57,58,59]. BL22 was associated with hemolytic uremic syndrome (HUS), a side effect not seen with LMB-2. Fortunately, this type of HUS, which occurred in 12% and 5% of HCL patients treated in phase 1 and 2, respectively, was fully reversible and did not require plasmapheresis for complete resolution, indicating that its mechanism is different and less serious than HUS due to a Shiga-like toxin [115]. 

## 11. Construction of Recombinant Immunotoxin Moxetumomab Pasudotox

To enhance the activity and specificity of BL22, the complementarity determining region 3 (CDR3) hypervariable domain of BL22 underwent hot-spot mutagenesis. Using phage display selection, a mutant of BL22 was isolated containing THW instead of SSY at positions 100, 100a, and 100b of the VH. This mutant has a 14-fold higher binding affinity to CD22 compared with BL22, mainly due to a slower off-rate [116]. This higher binding affinity causes an improvement in cytotoxicity toward both HCL and CLL. The new recombinant immunotoxin was initially named HA22, then CAT-8015, and finally, during clinical testing, moxetumomab pasudotox (Moxe). Preclinical studies included an animal study that showed improved antitumor activity of Moxe compared with BL22 [110]. With these data, phase 1 clinical testing of Moxe began in May of 2007, in relapsed/refractory HCL.

## 12. Moxe Phase 1 Clinical Results

The first 28 patients enrolled onto the phase I Moxe study in relapsed HCL included a dose-escalation cohort of 16 patients who received 5, 10, 20, 30, and 40 μg/kg every other day for three doses (QOD ×3) for each cycle with approximately 28-day cycles. Twelve patients received 50 μg/kg [60]. Patients had received 1–7 (median 2) prior courses of purine analog and 11 (46%) patients were refractory to their last course of purine analog defined as CR/PR <1 year. Seven had prior splenectomy and the other 21 had spleens with a median diameter of 150 mm. Patients received 2–16 (median four) cycles. Thirteen (46%) patients achieved CR, with an overall response rate (ORR) of 86%. Of the 13 CRs, 10 (77%) remained in CR at a median of 29 months of follow-up. The CR rate was not related to the number of prior courses of purine analog or to the duration of response to the last purine analog [60]. In contrast, CR rate was related to spleen status, in that patients with prior splenectomy (N = 7) had no CRs, while 13 (62%) of 21 with spleens present up to 325 mm in diameter had CR (*p* = 0.007). Our hypothesis for the lower CR rate in patients’ post-splenectomy is that these patients have more advanced disease since splenectomy resolves cytopenias, requiring higher tumor infiltration into the bone marrow to result in cytopenias sufficient to qualify for treatment. We therefore recommend that patients receive Moxe prior to splenectomy. We did not observe dose-limiting toxicities (DLTs). HUS was observed in two patients with Moxe, but it was only of moderate (grade 2) severity, limited to grade 1 thrombocytopenia and grade 1 creatinine elevations. These two patients included one at 30 and one at 50 μg/kg. Both patients fully recovered, and these mild laboratory abnormalities might not have been detected had the patients not been monitored carefully for HUS due to the experience with BL22. Few grade 3-4 toxicities were observed with Moxe, including grade 3–4 lymphopenia and leukopenia which probably represented treatment effects due to the targeting of malignant and normal CD22+ B-cells. Grade 1-2 toxicity included hypoalbuminemia, edema, weight gain, and proteinuria consistent with capillary leak syndrome (CLS). Hepatic enzyme elevation not associated with impaired hepatic function was also observed and even at grade 3 was not considered dose-limiting per protocol.

## 13. Expansion of the Phase 1 Trial of Moxe

To determine the clinical activity of Moxe at a fixed dose level, the phase I study was expanded to enroll 21 more patients at 50 μg/kg, for a total of 49 phase 1 patients and a total of 33 at 50 μg/kg, the peak dose level [64]. We did not observe additional cases of HUS in these additional 21 patients. With the 33 patients at 50 μg/Kg × 3 doses/cycle, we focused on MRD, assessed by bone marrow immunohistochemistry and blood and bone marrow aspirate flow cytometry. The detection limit was 0.002% by flow cytometry. Once achieving MRD-free CR, patients could receive two more cycles, called “consolidation” cycles. Figure 2 shows hematologic improvement and eradication of circulating HCL cells in the first patient treated at 50 μg/Kg QOD ×3. This patient has remained in CR and is still MRD free at the 10.5-year time point after CR was first documented. Of 33 patients receiving 50 μg/kg QOD ×3, the ORR was 88% and 21 (64%) achieved CR. Twenty patients achieving CR were evaluable for MRD by all tests including bone marrow aspirate flow cytometry. Of these, 11 (55%) achieved MRD-free CR and only one of these relapsed. The other 10 patients with MRD-free CR remained in CR with median CR duration up to 72 months, median 42 months. Of nine patients with MRD+ CR, eight relapsed, making median CR duration significantly prolonged for MRD-free vs. MRD+ CR (not reached vs. 13.5 months *p* < 0.0001). This was, to our knowledge, the first report that eradication of MRD in HCL is associated with longer CR duration. The duration of MRD-positive CR in this trial, which was for HCL patients in ≥third line, was shorter than the duration of MRD-positive CR after first-line HCL treatment with cladribine [67]. We therefore believe it is particularly critical to eliminate MRD in multiply relapsed HCL. 

## 14. Pharmacokinetics of Moxe by Bioassay

Since the toxin in Moxe is bacterial, we expected immunogenicity, but were surprised that in the phase 1 trial only 1 of 28 evaluable patients made high levels of neutralizing antibodies after the first cycle of Moxe [60]. This allowed most of the patients to be retreated to optimize response. The assay of neutralizing antibodies on the phase 1 trial involved incubating 200 and 1000 ng/mL mixtures of Moxe containing 90% patient serum for 15 min at 37 °C and comparing cytotoxic activity of these mixtures to those with the same concentration of Moxe incubated with albumin-containing saline. The cytotoxicity assay involved incubating dilutions of the mixtures with aliquots of CD22+ Raji cells and measuring inhibition of protein synthesis. To determine pharmacokinetics, dilutions of patient plasma after treatment were incubated with Raji cells and cytotoxicity compared to a standard curve made with purified Moxe. At the highest phase 1 dose level, 29 patients were evaluable by plasma for pharmacokinetic assays. These assays usually showed relatively low levels of Moxe during the first dose, due to the high CD22 density on tumor cells causing a CD22-sink [60]. Moxe achieved rapid tumor reductions during cycle 1 between the first and third doses, leading to a significant increase in peak level and area under the curve (AUC), and a decrease in volume of distribution (Vd) and clearance. On subsequent cycles, tumor burden remained diminished and the differences between the first (day 1) and third (day 5) doses were less pronounced. 

## 15. Antidrug Antibodies and Moxe Pharmacokinetics

These Moxe phase 1 patients were also evaluated for antidrug antibodies (ADA), and 15 evaluable patients had >50% neutralization of 200 ng/mL of Moxe prior to cycles 2 (*n* = 3), 3 (*n* = 5), 4 (*n* = 4), 5 (*n* = 2), and 6 (*n* = 1). This level of ADA disqualified the patients from subsequent cycles, but they received that “last” cycle since the ADA levels were not known until after cycle completion. In nearly all these extra cycles, peak levels and AUC increased and Vd and clearances decreased between days 1 and 5 [64]. This indicates that repeated doses of Moxe overcame neutralizing antibodies and that even secondary immunogenicity was insufficient to prevent cytotoxic and potentially therapeutic plasma levels of Moxe. Of these 15 patients, 9 went on to achieve CR. These data, together with the lack of toxicity during ADA development, justified repeat cycles of Moxe regardless of ADA, and hence justified not screening for ADA during phase 3 for eligibility or retreatment. 

## 16. Pivotal Phase 3 Testing of Moxe, Trial Design

For the pivotal trial of Moxe that led to its FDA approval, Moxe was evaluated in a single-arm international multicenter phase 3 trial at 32 centers in 14 countries in patients with relapsed/refractory HCL [65]. Eligibility included at least two prior treatments including at least one course of purine analog, and additional treatment with either a second course of purine analog or rituximab or BRAF inhibitor. Patients required at least one cytopenia or symptomatic splenomegaly. Creatinine had to be ≤1.5 mg/dL or the estimated creatinine clearance had to be ≥60 mL/min. Exclusions included brain metastases and uncontrolled infection or organ failure. A total of 80 patients were enrolled and received Moxe at 40 μg/kg on days 1, 3, and 5 in 28-day cycles for up to six cycles. Patients had to stop earlier than six cycles in the event of progressive disease, unacceptable toxicity, or documentation of MRD-free CR. The 40 μg/kg dose level during phase 3 was similar in potency to the 50 μg/kg dose level of phase 1 Moxe due to improvements in production and purity. Since patients had to stop retreatment after documentation of MRD-free CR, those who desired to receive the maximum number of cycles could not be restaged until after six cycles were complete. Thus, in the phase 3 trial, it was not possible to determine the number of consolidation cycles after MRD-negative CR. It should be noted that while MRD was assessed by bone marrow biopsy immunohistochemistry through a central lab, the assessment of MRD-negative CR for purposes of stopping retreatment was performed at each site, where the sensitivity of MRD tests likely differed. Since bone marrow aspirate flow cytometry is more sensitive than bone marrow biopsy immunohistochemistry [67], centers utilizing only the latter MRD test might have a higher rate of MRD-negative CR. The primary endpoint was to determine the percent of patients who could achieve CR, which required reversal of cytopenias to the level of a hematologic remission (HR), and then document absence of recurrent cytopenias (continued HR) over 180 days. 

## 17. Moxe Phase 3 Pivotal Trial Results

Phase 1 and 3 results are compared in Table 2. A total of 89 patients were tested for eligibility and 80 patients were enrolled. Patient ages were 34–84 (median 60) years, patients had 2–11 (median 3) prior lines of therapy, and half of the patients had ≥3 prior lines, evidence that this was a heavily pretreated population [65]. Of the 80 patients, 75% had prior rituximab, 29% had prior rituximab combined with purine analog, and 18% had a BRAF inhibitor. Fifty (63%) patients completed all six cycles and 12 (15%) received fewer than six cycles due to MRD-free CR. Another 12 (15%) stopped early due to an adverse event and this was treatment-related in eight (10%) patients. The median duration of follow-up was 16.7 months as of data cutoff in May of 2017. The CR rate was 41% (33 of 80) for an ORR of 75%. Durable CR over 180 days was achieved in 24 (30%) of the patients at a median follow-up of 16.7 months. Thus, there were nine (11%) patients who achieved CR but not durable CR. However, only two of these nine had recurrent cytopenias during the 180-day period of follow-up. Five of these nine patients actually did have durable CR, but the CR began later than the end of treatment time point, so durable CR could not be documented at the time of the report. In the remaining two cases, patients decided to travel and were not able to complete the monthly blood counts over six months. Among the 33 (41%) who achieved CR, 27 (82%) were negative for MRD by immunohistochemistry. Of six patients with MRD-positive CR, the median CR duration was 5.9 months vs. not reached in the 27 patients with MRD-free CR [65]. Regarding safety, grade 3–4 treatment-related adverse events occurred in 13 (16%) of the 80 patients including lymphopenia (30%), HUS (5%), infection (2.5%), and CLS (2.5%). All HUS and CLS events were reversible. Three patients (3.8%) died of infection including pneumonia and sepsis, none of these related to Moxe. Based on the phase 1 and 3 trials, Moxe was approved by the FDA on 13 September 2018, under the name Lumoxiti, for patients with relapsed/refractory HCL who had received at least two prior systemic therapies, including treatment with a purine analog.

## 18. Mechanism and Prevention of HUS and CLS with Moxe

In the phase 3 trial, the incidence of CLS and HUS of all grades was 5% and 7.5%, respectively [65]. In our experience with HUS from BL22 (seven cases) and Moxe (six cases) in HCL, HUS presented on day 7 or 8, i.e., 2–3 days after the last dose of Moxe given days 1, 3, and 5, and never presented after day 8 [57,58,59,60,64,65]. For that reason, we recommend obtaining labs on day 8, looking for increases in creatinine, lactate dehydrogenase (LDH), and bilirubin, and decreases in platelets, hemoglobin, and haptoglobin. Abnormalities might be limited to grade 1 thrombocytopenia and creatinine elevation, but either of these alone on day 8 would not indicate HUS. Although the mechanism of immunotoxin-induced HUS is unknown, it is known that endothelial damage exposes ultra-large multimers of von Willebrand’s factor under the endothelial surface which can constitute a site for platelet aggregation and thrombin formation [115]. CLS, which involves movement of fluid and protein out of the blood vessels, can result in intravascular volume depletion, an increase in Moxe concentration in the glomerular capillaries, and hence increase the risk of HUS. While CLS is a common effect of immunotoxins targeting a variety of antigens [55,56,117,118,119,120,121,122,123], HUS has only been reported from those targeting CD22 [57,58,59,60,64,65,80,81,124,125,126]. Immunohistochemistry studies showed no evidence that CD22 is expressed on glomerular endothelial cells which would be targeted specifically by Moxe. If that were the case, we could expect HUS to occur at lower doses of Moxe than of BL22, since Moxe binds with 14-fold improved affinity to CD22 [116]. Rather, we believe the Fv fragment of the Mab-targeting CD22 interacts, perhaps weakly, with a non-CD22 antigen present in the glomerulus. To prevent HUS, we advocate keeping the concentration of Moxe in the glomerulus as low as possible by ensuring adequate hydration. While excessive IV fluid can worsen CLS, leading to systemic and pulmonary edema, we find that oral hydration is highly effective without worsening CLS. Since CLS is constant, we recommend patients drink an average of about 250 mL (1 cup) of water every hour during days 1–8 of each cycle of Moxe, and not go more than 2–3 h at night without drinking. While this requires patient education and follow-up by the treating team [127], we find this is well tolerated, and in most patients does not cause hyponatremia. Some side effects occurring 6–24 h after Moxe, including headache, nausea, and fever, can lead to hypovolemia or decreased water intake, and these can be rapidly eliminated with dexamethasone 4 mg orally. Most patients need oral dexamethasone after one or two of the three doses of Moxe, and some patients need none. Before using precautions of oral hydration and dexamethasone, three of our first nine patients enrolled on the phase 3 trial had grade 3 HUS, and after the precautions, only 1 of the next 17 patients had a grade 1 HUS (*p* = 0.032 for grade 3 HUS). While this was not a randomized comparison, we are using these precautions to improve the safety of Moxe and allow patients to receive a full 6 cycles to achieve MRD-free CR.

## 19. ADA and Pharmacokinetics of Moxe during Phase 3 Testing

To determine ADA and plasma levels during phase 3, ELISA assays rather than bioassays were utilized. Higher levels and more frequent positivity for ADA by ELISA were expected compared with ADA by bioassay. This is because ELISA is more sensitive and detects both neutralizing and non-neutralizing antibodies binding to Moxe, while the bioassay only detects neutralizing antibodies. ELISA assays detected a 59% rate of ADA at baseline and 88% prevalence of ADA at least at one time point and increased as expected following treatment. An ELISA test was developed to determine ADA specificity for different domains of Moxe [128]. ADA at baseline was always associated with antibodies binding to the toxin domains of Moxe and these anti-toxin antibodies increased with treatment [128]. This was expected based the bacterial origin of PE38. While the median ADA titer increased with retreatment in all response groups, patients with CR had the lowest levels of ADA, followed by those achieving PR, stable disease, and progressive disease as best response, although statistical significance was not reported [65]. For 49 phase 1 and 74 phase 3 patients, pharmacokinetic data determined by ELISA were pooled to correlate with patient characteristics, disease burden, prior treatment, and ADA [129]. The model showed that the linear clearance from the central compartment after the first dose of Moxe in cycle 1 was 24.7 L/h, compared with 3.75 L/h for subsequent doses. Patients with ADA titers >10,240 had 4-fold higher clearances than those with lower levels of ADA. While exposures higher than the median correlated with a higher response rate, CLS, and creatinine rises during phase 3, clinical benefit was observed even in patients with low exposure or high ADA. This showed that there is clinical activity of Moxe despite immunogenicity and resulting increased clearances [129]. 

## 20. Further Development of Moxe for HCL

To increase the rate of patients achieving MRD-free CR with Moxe, we are now testing Moxe in combination with rituximab. The rationale for this approach is two-fold. First, since rituximab depletes circulating B-cells to undetectable levels for about six months after the last dose, it may prevent ADA and improve drug exposure. Secondly, rituximab should reduce tumor burden, enabling Moxe to reach more tumor cells and achieve CR earlier. Even if rituximab does not decrease ADA and ADA increases after a certain number of cycles of Moxe, a faster response to Moxe due to the cell killing action of rituximab will increase the likelihood of MRD-free CR. It was reported that rituximab failed to prevent immunogenicity of LMB-1-targeting solid tumors [130]. Despite those results, rituximab may prevent ADA in patients with HCL since patients with B-cell malignancies have lower humoral immunity than patients with solid tumors, and since Moxe lacks the murine IgG constant domains of LMB-1 and should be less immunogenic. In the ongoing trial of Moxe-rituximab, the rituximab is begun on day-2, and Moxe on days 1, 3, and 5. The three-day interval between rituximab and Moxe on cycle 1 is designed to give rituximab enough time to reduce normal B-cells and the HCL burden. On subsequent cycles, spaced 28 days apart, rituximab is begun just prior to Moxe on day 1. Although rituximab is marginally effective in relapsed HCL as a single agent [131], we recently reported that its addition to first line cladribine results in a marked improvement in MRD-free CR, from 24% to 97% at six months [67]. In comparing Moxe-rituximab to cladribine-rituximab (CDAR), albeit across trials, it is notable that Moxe as a single-agent has a higher MRD-free CR rate than cladribine [64]. Vemurafenib, which did not eradicate MRD by immunohistochemistry as a single-agent [68], achieved an MRD-free CR rate of 65% in relapsed HCL when combined with rituximab [132], and the vemurafenib-obinutuzumab combination is currently being tested in first-line. We believe the Moxe-rituximab regimen, or possibly Moxe-obinutuzumab, may have potential in the future as a chemotherapy-free first-line treatment of HCL. 

## Figures and Tables

**Figure 1 biomolecules-10-01140-f001:**
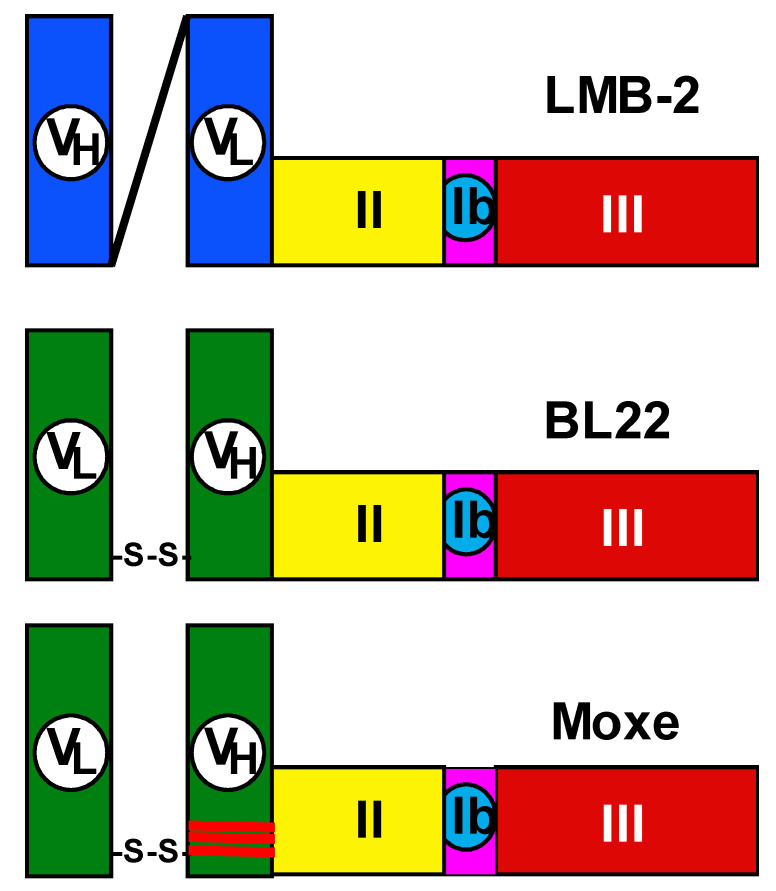
Recombinant immunotoxins targeting hematologic malignancies. LMB-2 development began in 1989 and led to the anti-CD25 single-chain recombinant immunotoxin LMB-2, also called anti-Tac(Fv)-PE38 [99,100,101]. The variable domains of the anti-CD25 Mab anti-Tac are connected by a peptide linker (GGGGS)_3_, and a shorter peptide connects V_L_ with the 38 kDa truncated form of PE called PE38. LMB-2 entered clinical testing in July 1996. BL22 contains the variable domains of the RFB4 Mab [102] targeting CD22. V_H_ and V_L_ are disulfide-stabilized by converting Arg44 of V_H_ and Gly100 of V_L_ to cysteines and allowing the disulfide bond to form during redox renaturation refolding [103]. The carboxy terminus of VH is connected to PE38. BL22 began clinical testing in February 1999. Moxetumomab pasudotox (Moxe) is an affinity-matured form of BL22 containing 3 mutations in VH. Moxe began clinical testing in May 2007 and was approved by FDA in September 2018.

**Figure 2 biomolecules-10-01140-f002:**
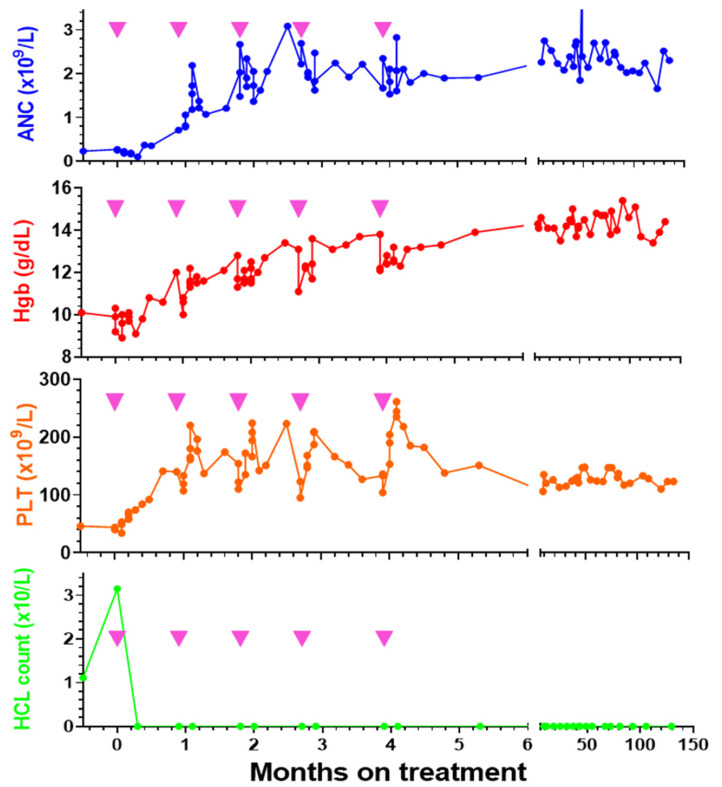
Patient HH17 was treated with Moxe 50 μg/Kg QOD ×3 for 5 cycles, with the cycles indicated by the inverted pink triangles. Minimal residual disease (MRD)-free complete remission (CR) was achieved prior to cycle 4, and the patient was still MRD-free by bone marrow immunohistochemistry and blood and bone marrow flow cytometry 129 months after treatment, 10.5 years in MRD-free CR.

**Table 1 biomolecules-10-01140-t001:** Criteria for response in hairy cell leukemia (HCL).

	Consensus Guidelines [26,27]	Variations
Complete remission (CR)		
CBC	ANC >1.5, Hgb >11, and Plt 100	Hgb ≥12 for males [62]
CBC duration	One resolved CBC is sufficient	Resolved CBCs ≥4 weeks [19,59,60,62,64,65,66]
HCL cells	Absent in blood and marrow by morphology	
Spleen	Absent on exam	Resolved by CT/MRI [65]
Lymph nodes	Not specified	Resolved to ≤2cm in short axis by imaging/exam [66]
CR, MRD-free		
Bone marrow	IHC Negative by CD20, DBA.44, or VE1	IHC and flow cytometry negative
Partial response (PR)		
CBC	Same as CR	≥50% improvement in ANC, Hgb, and Plt [21,55,56,57,58,59,60,62,63]
CBC duration	No minimum duration	Resolved CBC ≥4 weeks [19,59,60,62,64,65,66]
Blood HCL	Not specified	≥50% reduction [19,21,68]
Marrow HCL	≥50% reduction in infiltration	Avoid due to heterogeneity [64,65]
Organomegaly	≥50% reduction	Require CT/MRI imaging due to bias [65]
Progressive disease (PD)		
CBC	≥25% decline in ANC, Hgb, or Plt, due to HCL	
Symptoms	Increase in symptoms related to disease	
Organomegaly	≥25% increase in organomegaly	
Lymph nodes	Not specified	New lymph nodes or ≥25% increase in existing nodes [65,71]
Blood HCL	Not specified	≥50% increase in HCL cells or absolute lymphocytes [65,71]
Stable disease	Neither CR, PR, nor PD	

CBC, complete blood count; ANC, absolute neutrophil count; Hgb, hemoglobin.

**Table 2 biomolecules-10-01140-t002:** Results of moxetumomab pasudotox (Moxe) during phase 1 and 3 testing.

	Phase 1	Phase 3
Patients treated	49	80
Eligibility, clinical	Cytopenias or Symptomatic splenomegaly	Cytopenias or Symptomatic splenomegaly


Eligibility, ADA	Negative test for ADA	Testing not needed
Doses tested (μg/Kg QOD ×3)	5, 10, 20, and 30 (n = 3 each)	40 (n = 80)
	40 (n = 4), 50 (n = 33)	
Patient ages	40–77 (median 57)	34–84 (median 60)
Male-Female ratio	41:8	68:12
HCLv	2 (4%)	3 (3.8%)
Prior purine analog courses		
1	4 (8%)	10 (13%)
2	24 (49%)	30 (38%)
≥3	21 (43%)	40 (50%)
Prior rituximab	30 (61%)	60 (75%)
Prior splenectomy	8 (16%)	5 (6%)
ORR	42 (86%)	60 (75%)
CR rate	28 (57%)	33 (41%)
Toxicity, dose level assessed	50 μg/Kg × 3 (n = 33)	40 μg/Kg × 3 (n = 80)
Grades 1–4, 3–4		
Hemolytic uremic syndrome	2 (4%), 0 (0%)	7 (9%), 4 (5%)
Capillary leak syndrome	8 (16%), 0 (0%)	7 (9%), 2 (3%)

ADA, antidrug antibodies; ORR, overall response rate. The 50 μg/Kg phase 1 and 40 μg/Kg phase 3 doses had similar potency due to improvements in the manufacturing and purity of the phase 3 lot.

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
