# Peer review of "Development of Recombinant Immunotoxins for Hairy Cell Leukemia"

_biomolecules, 2020, doi:10.3390/biom10081140_

Round 1

Reviewer 1 Report

In the following "Development of recombinant immunotoxins for hairy cell leukemia", Dr. Robert Kreitman and Dr. Ira Pastan reviewed the immunotoxins rationale usage for Hairy Cell Leukemia, a lymphoproliferative disorder with great clinical unmet needs. Authors have focused on Moxetumomab pseudotox (Moxe), a promising agent for HCL relapsed or refractory. The review is of interest for readers of Biomolecules and suitable for publication. Some minors comments are listed below: 1) Please review the manuscript, there are some typos across the text. Another aspect important to review is the abbreviations not listed, such as BMA (Bone arrow Aspiration?) 2) References format seems inconsistent in some parts (e.g. Plant vs Bacterial Toxins); 3) This reviewer would like to suggest expanding the MRD definitions and methods limitations and applications; 4) I would like to suggest adding a table with criteria for response; 5) Despite the promising effects of Moxe, seems important to me to expand the rationale explained in the introduction of this agent

Author Response

Thank you for these comments.  We fixed typo’s, and the technical problems with references.  As suggested, we significantly added to the MRD section regarding definitions of MRD.  As suggested we constructed a detailed and referenced table for response criteria.  We also expanded the rationale for development of anti-CD22 recombinant immunotoxins for HCL and added it to the beginning of the paragraph with that title. 

Reviewer 2 Report

This is an excellent comprehensive review from an established investigative group. An excellent example of translational molecular therapeutics...all the way to the clinic and FDA approval. 

All of my concerns are minor.

The manuscript does need to be edited.

For example, line 215, 241,250, 302 lists g/kg. I think they mean ug/kg. 

Some of the sections have a bulky sentence structure. For example, line 223-225. 

A few sentences should address, the choice of CD22 as a target over CD19 since CD19 is such a popular target for B cell malignancies.

The choice of font for Table 3 is poor. It looks like it was inserted heater-skelter. 

The first part of the review is "text dense" and it would be very helpful to have a Figure illustrating the drug structure and  the evolution of their drug development LMB-2 to BL2 to  H22 Mox.

Author Response

In addition to editing the noted places, we added a reference for improved internalization of CD22 compared to CD19, as a rationale for CD22 rather than CD19 immunotoxin.  The font for table II was reduced to allow more space between columns, and the table was put onto 1 page.  .As suggested, we constructed a figure explaining the evolution from LMB-2 to BL22 to Moxe.